# An Algorithm to Learn Polytree Networks with Hidden Nodes

**Firoozeh Sepehr**
Department of EECS
University of Tennessee Knoxville
1520 Middle Dr, Knoxville, TN 37996
dawn@utk.edu

**Donatello Materassi**
Department of EECS
University of Tennessee Knoxville
1520 Middle Dr, Knoxville, TN 37996
dmateras@utk.edu

## Abstract

Ancestral graphs are a prevalent mathematical tool to take into account latent (hidden) variables in a probabilistic graphical model. In ancestral graph representations, the nodes are only the observed (manifest) variables and the notion of $m$-separation fully characterizes the conditional independence relations among such variables, bypassing the need to explicitly consider latent variables. However, ancestral graph models do not necessarily represent the actual causal structure of the model, and do not contain information about, for example, the precise number and location of the hidden variables. Being able to detect the presence of latent variables while also inferring their precise location within the actual causal structure model is a more challenging task that provides more information about the actual causal relationships among all the model variables, including the latent ones. In this article, we develop an algorithm to exactly recover graphical models of random variables with underlying polytree structures when the latent nodes satisfy specific degree conditions. Therefore, this article proposes an approach for the full identification of hidden variables in a polytree. We also show that the algorithm is complete in the sense that when such degree conditions are not met, there exists another polytree with fewer number of latent nodes satisfying the degree conditions and entailing the same independence relations among the observed variables, making it indistinguishable from the actual polytree.

## 1 Introduction

The presence of unmeasured variables is a fundamental challenge in discovery of causal relationships [1, 2, 3]. When the causal diagram is a Directed Acyclic Graph (DAG) with unmeasured variables, a common approach is to use ancestral graphs to describe the independence relations among the measured variables [2]. The main advantage of ancestral graphs is that they involve only the measured variables and successfully encode all their conditional independence relations via $m$-separation. Furthermore, complete algorithms have been devised to obtain ancestral graphs from observational data, e.g., the work in [3]. However, recovering the actual structure of the original DAG is something that ancestral graphs somehow circumvent. For example, it might be known that the actual causal diagram has a polytree structure including the hidden nodes, but the ancestral graph associated with the measured variables might not even be a polytree [4]. Instead, the recovery of causal diagrams including the location of their hidden variables is a very challenging task and algorithmic solutions are available only for specific scenarios [5, 6, 7, 8]. For example, in the case of specific distributions (i.e., Gaussian and Binomial) when the causal diagram is known to be a rooted tree, the problem has been solved by exploiting the additivity of a metric along the paths of the tree [6, 7, 8, 9]. In the case of generic distributions, though, additive metrics might be too difficult to define or cannot be defined in general. Furthermore, rooted trees can be considered a rather limiting class of networks

since they represent probability distributions which can only be factorized according to second order conditional distributions [10].

This article makes a novel contribution towards the recovery of more general causal diagrams. Indeed, it provides an algorithm to learn causal diagrams making no assumptions on the underlying probability distribution, and considering polytree structures which can represent factorizations involving conditional distributions of arbitrarily high order. Furthermore, it is shown that a causal diagram with a polytree structure can be exactly recovered if and only if each hidden node satisfies the following conditions: (i) the node has at least two children; (ii) if the node has exactly one parent, such a parent is not hidden; (iii) the node has at least degree 3, or each of its two children has at least another parent. The provided algorithm recovers every polytree structure with hidden nodes satisfying these conditions, and, remarkably, makes use only of third order statistics. If the degree conditions are not satisfied, then it is shown that there exists another polytree with fewer number of hidden random variables which entails the same independence relations among the observed variables. Indeed, in this case, when no additional information/observations are provided, no test can be constructed to determine the true structure. Another main advantage of this proposed approach lies in the fact that it follows a form of Occam's razor principle since in the case where the degree conditions on the hidden nodes are not met, then a polytree with minimal number of hidden nodes is selected. We find this property quite relevant in application scenarios since Occam's razor is arguably one of the cardinal principles in all sciences.

## 2 Preliminaries, Assumptions and Problem Definition

In order to formulate our problem, we first introduce a generalization of the notions of directed and undirected graphs (see for example [11, 12]) which also considers a partition of the set of nodes into visible and hidden nodes.

**Definition 1** (Latent partially directed graph). *A latent partially directed graph $\bar{G}_\ell$ is a 4-ple $(V, L, E, \vec{E})$ where*

- *the disjoint sets $V$ and $L$ are named the set of visible nodes and the set of hidden nodes,*
- *the set $E$ is the set of undirected edges containing unordered pairs of $(V \cup L) \times (V \cup L)$,*
- *the set $\vec{E}$ is the set of directed edges containing ordered pairs of $(V \cup L) \times (V \cup L)$.*

*We denote the unordered pair of two elements $y_i, y_j \in V \cup L$ as $y_i - y_j$, and the ordered pair of $y_i, y_j$ (when $y_i$ precedes $y_j$) as $y_i \rightarrow y_j$. In a latent partially directed graph the sets $E$ and $\vec{E}$ do not share any edges. Namely, $y_i - y_j \in E$ implies that both $y_i \rightarrow y_j$ and $y_j \rightarrow y_i$ are not in $\vec{E}$.* □

A latent partially directed graph is a fully undirected graph when $\vec{E} = \emptyset$, and we simplify the notation by writing $G_\ell = (V, L, E)$. Similarly, when $E = \emptyset$, we have a fully directed graph, and we denote it by $\vec{G}_\ell = (V, L, \vec{E})$. Furthermore, if we drop the distinction between visible and hidden nodes and consider $V \cup L$ as the set of nodes, we recover the standard notions of undirected and directed graphs. Thus, latent partially directed graphs inherit, in a natural way, all notions associated with standard graphs (e.g., path, degree, neighbor, etc., see for example [11]). In the scope of this article, we denote degree, outdegree, indegree, children, parents, descendants and ancestors of a node $y$ in graph $\vec{G}$ using $\deg_{\vec{G}}(y)$, $\deg_{\vec{G}}^+(y)$, $\deg_{\vec{G}}^-(y)$, $\mathrm{ch}_{\vec{G}}(y)$, $\mathrm{pa}_{\vec{G}}(y)$, $\mathrm{de}_{\vec{G}}(y)$ and $\mathrm{an}_{\vec{G}}(y)$, respectively (see [11, 12] for precise definitions). Furthermore, the notion of restriction of a graph to a subset of nodes follows immediately.

**Definition 2** (Restriction of a latent partially directed graph). *The restriction of a latent partially directed graph $\bar{G}_\ell = (V, L, E, \vec{E})$ with respect to a set of nodes $A \subseteq V \cup L$ is the latent partially directed graph obtained by considering only the nodes in $A$ and the edges linking pairs of nodes which are both in $A$.* □

Moreover, a latent partially directed graph is called a latent partially directed tree when there exists exactly one path connecting any pair of nodes.

**Definition 3** (Latent partially directed tree). *A latent partially directed tree $\vec{P}_\ell$ is a latent partially directed graph $\bar{G}_\ell = (V, L, E, \vec{E})$ where every pair of nodes $y_i, y_j \in V \cup L$ is connected by exactly one path.*

Trivially, latent partially directed trees generalize the notions of undirected trees and polytrees (directed trees) [13]. In a latent partially directed tree, we define a hidden cluster as a group of hidden nodes that are connected to each other via a path constituted exclusively of hidden nodes.

**Definition 4** (Hidden cluster). *A hidden cluster in a latent partially directed tree $\vec{P}_\ell = (V, L, E, \vec{E})$ is a set $C \subseteq L$ such that for each distinct pair of nodes $y_i, y_j \in C$ the unique path connecting them contains only nodes in $C$ and no node in $C$ is linked to a node which is in $L \setminus C$.* □

Observe that each node in a hidden cluster has neighbors which are either visible or hidden nodes of the same cluster. Figure 1 (a) depicts a latent directed tree (or a latent polytree) and its hidden clusters $C_1$ and $C_2$ highlighted by the dotted lines.

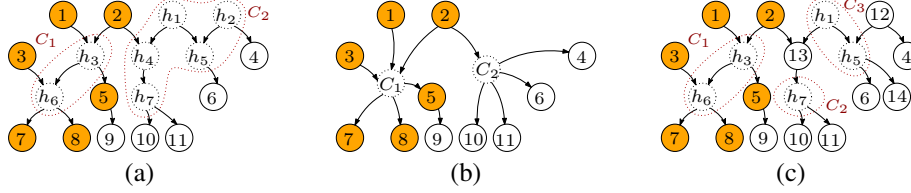

$$(a) \qquad\qquad (b) \qquad\qquad (c)$$

Figure 1: A hidden polytree (a) and its collapsed hidden polytree (b), a minimal hidden polytree (c).

Furthermore, we introduce the set of (visible) neighbors of a hidden cluster, its closure and its degree.

**Definition 5** (Neighbors, closure, and degree of a hidden cluster). *In a latent partially directed tree, the set of all visible nodes linked to any of the nodes of a hidden cluster $C$ is the set of neighbors of $C$ and is denoted by $N(C)$. We define the degree of the hidden cluster as $|N(C)|$, namely, the number of neighbors of the cluster. We refer to the restriction of a latent polytree to a hidden cluster and its neighbors as the closure of the hidden cluster.* □

Observe that the neighbors of $C_1$ are shaded with orange color in Figure 1 (a). We also remind the notion of a root node and define the notion of a root of a hidden cluster.

**Definition 6** (Root of a latent polytree, and root of a hidden cluster in a latent polytree). *In a latent polytree $\vec{P}_\ell = (V, L, \vec{E})$, a root is a node $y_r \in V \cup L$ with indegree equal to zero. Also, we define any root of the restriction of the polytree to one of its hidden clusters as the root of the hidden cluster.* □

For example, in Figure 1 (a), node $y_1$ is a root of the latent polytree and node $y_{h_3}$ is a root of the hidden cluster $C_1$. In this article, we make extensive use of the restriction of a polytree to the descendants of one of its roots. We define such a restriction as the rooted subtree of the polytree associated with that root. Additionally, given a latent partially directed tree, we define its *collapsed representation* by replacing each hidden cluster with a single hidden node. The formal definition is as follows and Figure 1 (b) depicts the collapsed representation of the latent polytree of Figure 1 (a).

**Definition 7** (Collapsed representation). *We define the collapsed representation of $\vec{P}_\ell = (V, L, E, \vec{E})$ as the latent partially directed tree $\vec{P}_c = (V, L_c, E_c, \vec{E}_c)$ where $n_c$ is the number of hidden clusters $C_1, ..., C_{n_c}$, and $L_c := C_1 \cup ... \cup C_{n_c}$, and*

$$E_c := \{y_i - y_j \in E \mid y_i, y_j \in V\} \cup \{y_i - C_k \mid \exists y_j \in C_k, y_i - y_j \in E\} \cup \{C_k - y_j \mid \exists y_i \in C_k, y_i - y_j \in E\}$$

$$\vec{E}_c := \{y_i \to y_j \in \vec{E} \mid y_i, y_j \in V\} \cup \{y_i \to C_k \mid \exists y_j \in C_k, y_i \to y_j \in \vec{E}\} \cup \{C_k \to y_j \mid \exists y_i \in C_k, y_i \to y_j \in \vec{E}\}.$$ □

In this article, we show the cases where graphical models with polytree structures can be recovered from the independence relations involving only visible nodes. Specifically, we assume that a polytree is a perfect map (see [14, 12]) for a probabilistic model defined over the variables $V \cup L$ where $V$ and $L$ are disjoint sets. We find conditions under which it is possible to recover information about the perfect map of the probabilistic model considering only independence relations of the form $I(y_i, \emptyset, y_j)$ (read $y_i$ and $y_j$ are independent) and $I(y_i, y_k, y_j)$ (read $y_i$ and $y_j$ are conditionally independent given $y_k$) for all nodes $y_i, y_j, y_k \in V$. One of the fundamental requirements of solving this problem is that all hidden nodes need to satisfy certain degree conditions summarized in the following definition.

**Definition 8** (Minimal latent polytree). *A latent polytree $\vec{P}_\ell = (V, L, \vec{E})$ is minimal if every hidden node $y_h \in L$ satisfies one of the following conditions:*

- $\deg^+_{\vec{P}_\ell}(y_h) \geq 2$ *and* $\deg^-_{\vec{P}_\ell}(y_h) \geq 3$ *and if* $|\mathrm{pa}_{\vec{P}_\ell}(y_h)| = 1$, *then* $\mathrm{pa}_{\vec{P}_\ell}(y_h) \subseteq V$;
- $\deg^+_{\vec{P}_\ell}(y_h) = 2$ *and* $\deg^-_{\vec{P}_\ell}(y_h) = 0$ *and* $\deg^-_{\vec{P}_\ell}(y_{c_1}), \deg^-_{\vec{P}_\ell}(y_{c_2}) \geq 2$ *where* $\mathrm{ch}_{\vec{P}_\ell}(y_h) = \{y_{c_1}, y_{c_2}\}$. $\square$

Note that the nodes $y_{h_2}$, $y_{h_4}$, $y_{h_5}$, $y_{h_7}$ in Figure 1 (a) do not satisfy the minimality conditions and therefore the hidden polytree is not minimal. Instead, Figure 1 (c) shows a minimal latent polytree.

The algorithm we propose to recover the structure of a latent polytree can be decomposed in several tasks and the hidden nodes which are roots with outdegree equal to 2 and at least one visible child require to be dealt with in a special way in the last task of the algorithm. Therefore, we define the following two types of hidden nodes to make this distinction.

**Definition 9** (Type-I and type-II hidden nodes). *In a minimal latent polytree, we classify a hidden node $y_h$ as type-II when $\deg_{\vec{G}}(y_h) = 2$ with at least one visible child. All other hidden nodes are classified as type-I.*

In the minimal latent polytree of Figure 2 (a), the hidden nodes $y_{h_2}$ and $y_{h_3}$ are type-II hidden nodes, while all the other hidden nodes are type-I.

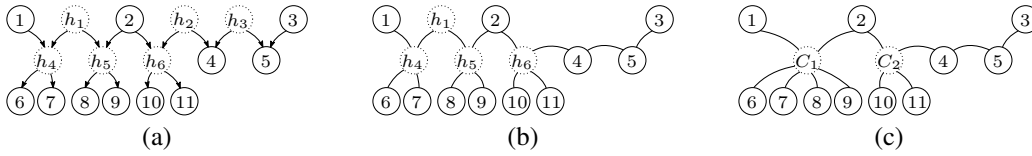

(a) (b) (c)

Figure 2: A minimal latent polytree $\vec{P}_\ell$ with type-II hidden nodes (a), its quasi-skeleton (b), and its collapsed quasi-skeleton (c).

We define the quasi-skeleton of a minimal latent polytree to deal with type-II hidden nodes separately.

**Definition 10** (Quasi-skeleton of a latent polytree). *In a minimal latent polytree $\vec{P}_\ell = (V, L, \vec{E})$, the quasi-skeleton of $\vec{P}_\ell$ is the undirected graph obtained by removing the orientation of all edges in $\vec{P}_\ell$, and removing all the type-II hidden nodes and then linking its two children together.* $\square$

In Figure 2 (b), we have the quasi-skeleton of the polytree of Figure 2 (a). Observe that we can easily define the collapsed representation of a quasi-skeleton of a latent polytree by finding the quasi-skeleton first and then finding its collapsed representation as in Figure 2 (c).

As it is well known in the theory of graphical models, in the general case, from a set of conditional independence statements (formally, a semi-graphoid) faithful to a Directed Acyclic Graph (DAG), it is not possible to recover the full DAG [15, 1]. What can be recovered for sure is the pattern of the DAG, namely the skeleton and the v-structures (i.e., $y_i \rightarrow y_k \leftarrow y_j$) of the DAG [15, 1]. In this article, we show that, similarly, in the case of a minimal latent polytree, we are able to recover the pattern of the polytree from the independence statements involving only the visible variables.

**Definition 11** (Pattern of a polytree). *Let $\vec{P} = (N, \vec{E})$ be a polytree. The pattern of $\vec{P}$ is a partially directed graph where the orientation of all the v-structures (i.e., $y_i \rightarrow y_k \leftarrow y_j$) are known and as many as the remaining undirected edges are oriented in such a way that the other alternative orientation would result in a v-structure.* $\square$

Now we have all the necessary tools to formulate the problem.

**Problem Formulation.** Assume a semi-graphoid defined over a set of variables $V \cup L$. Let the latent polytree $\vec{P}_\ell = (V, L, \vec{E})$ be faithful to the semi-graphoid and assume that the nodes in $L$ satisfy the minimality conditions. Recover the pattern of $\vec{P}_\ell$ from conditional independence relations involving only nodes in $V$.

**Remark 12.** *The proposed solution makes only use of the conditional independence relations of the form $\mathcal{I}(y_i, \emptyset, y_j)$ and $\mathcal{I}(y_i, y_k, y_j)$ for all $y_i, y_j, y_k \in V$.*

# 3 An Algorithm to Reconstruct Minimal Hidden Polytrees

Our algorithm for learning the pattern of a minimal latent polytree is made of the following 5 tasks:

1. Using the independence statements involving the visible nodes, determine the number of rooted subtrees in the latent polytree and their respective sets of visible nodes;
2. Given all the visible nodes belonging to each rooted subtree, determine the collapsed quasi-skeleton of each rooted subtree;
3. Merge the overlapping hidden clusters in the collapsed quasi-skeleton of each rooted subtree to obtain the collapsed quasi-skeleton of the latent polytree;
4. Determine the quasi-skeleton of the latent polytree from the collapsed quasi-skeleton of the latent polytree (recover type-I hidden nodes);
5. Obtain the pattern of the latent polytree from the recovered quasi-skeleton of the latent polytree (recover type-II hidden nodes and edge orientations).

Figure 3 shows the stage of the recovery of the polytree structure at the end of each task. The following subsections provide more details about each task, but the most technical results are in the Supplemental Material. We stress that the first two tasks mostly leverage previous work about rooted trees and the main novelty of this article lies in tasks 3, 4 and 5.

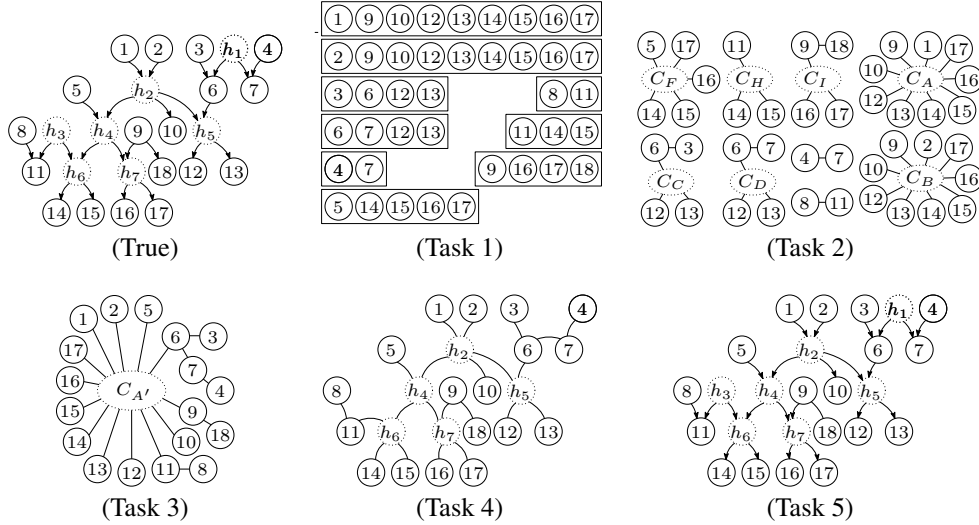

Figure 3: The actual minimal latent polytree (True); the lists of visible nodes for each rooted subtree (Task 1), collapsed quasi-skeletons of the rooted subtrees (Task 2), merging of the overlapping hidden clusters (Task 3), detection of type-I hidden nodes (Task 4), detection of type-II hidden nodes along with orientation of the edges to obtain the pattern of the hidden polytree (Task 5). Observe that the full polytree is not recovered at the end of task 5 since the edge $y_9 - y_{18}$ is left undirected.

## 3.1 Task 1: Determine the visible nodes of each rooted subtree

This first task can be performed by the Pairwise-Finite Distance Algorithm (PFDA), presented in [16] and reported in the Supplementary Material as Algorithm 4. As shown in [16], PFDA takes as input the set of visible nodes of a latent polytree and outputs sets of visible nodes with the property that each set corresponds to the visible descendants of a root of the latent polytree, when the polytree is minimal. In the following theorem, we show that the output of PFDA applied to the independence statements is the same as described above. See Supplementary Material for the proof of this theorem.

**Theorem 13.** *Consider a latent polytree $\vec{P}_\ell = (V, L, \vec{E})$ faithful to a probabilistic model. Assume that the hidden nodes in $L$ satisfy the minimality conditions. Then PFDA, applied to the independence statements of the probabilistic model with the form $\mathcal{I}(y_i, \emptyset, y_j)$ for all $y_i, y_j \in V$, outputs a collection of sets, such that each of them is given by all the visible descendants of a root of $\vec{P}_\ell$.*

## 3.2 Task 2: Determine the collapsed quasi-skeleton of each rooted subtree

The second task is performed by the Reconstruction Algorithm for Latent Rooted Trees in [17]. We report it as Algorithm 5 in the Supplementary Material for completeness. The input of this algorithm is the set $V_r$ of the visible nodes belonging to a rooted subtree $T_r$ and independence relations of the form $\mathcal{I}(y_i, y_k, y_j)$ or $\neg \mathcal{I}(y_i, y_k, y_j)$ for distinct $y_i, y_j, y_k \in V_r$. Its output is the collapsed quasi-skeleton of $T_r$. Thus, we can call this algorithm on all of the sets of visible nodes $V_1, ..., V_{n_r}$ where $n_r$ is the number of roots, obtained from Task 1, and find the collapsed quasi-skeletons of all the rooted subtrees of the latent polytree. This result is formalized in the following theorem. See Supplementary Material for the proof of this theorem.

**Theorem 14.** *Let $\vec{P}_\ell = (V, L, \vec{E})$ be a minimal latent polytree. Consider a root $y_r$ of $\vec{P}_\ell$ and let $V_r = V \cap \mathrm{de}_{\vec{P}_\ell}(y_r)$. The output of Reconstruction Algorithm for Latent Rooted Trees applied to $V_r$ is the collapsed quasi-skeleton of the rooted subtree with root node $y_r$.*

## 3.3 Task 3: Merge the overlapping hidden clusters of the collapsed rooted trees

By applying the Reconstruction Algorithm for Latent Rooted Trees on each set of visible nodes in the same rooted tree, we have, as an output, the collapsed quasi-skeletons of all rooted subtrees in the original hidden polytree. In the general case, some hidden clusters in the collapsed quasi-skeleton of the rooted subtrees might overlap, namely, they might share some hidden nodes in the original hidden polytree. The following theorem provides a test on the sets of visible nodes of the rooted subtrees in a minimal latent polytree to determine if two hidden clusters in two distinct collapsed quasi-skeletons of two rooted subtrees belong to the same cluster in the collapsed quasi-skeleton of the polytree. See Supplementary Material for the proof of this theorem.

**Theorem 15.** *Consider a minimal latent polytree $\vec{P}_\ell$. Let $C_1$ and $C_2$ be two distinct hidden clusters in the collapsed quasi-skeletons of two rooted subtrees of $\vec{P}_\ell$. If the set of neighbors of $C_1$ and the set of neighbors of $C_2$ share at least a pair of visible nodes, i.e., $|N(C_1) \cap N(C_2)| \geq 2$, then the nodes in $C_1$ and $C_2$ belong to the same hidden cluster in the collapsed quasi-skeleton of $\vec{P}_\ell$.*

This theorem is the enabling result for the Hidden Cluster Merging Algorithm (HCMA), presented in Algorithm 1, which merges all the collapsed quasi-skeletons associated with the individual rooted subtrees, obtained from Task 2, into the collapsed quasi-skeleton of the polytree. This algorithm starts with the collapsed quasi-skeleton of the rooted subtrees, then finds pairs of clusters that overlap by testing if they share at least one pair of visible neighbors (see Theorem 15), and then merges the overlapping pairs. This procedure is repeated until no clusters are merged anymore.

---

**Algorithm 1** Hidden Cluster Merging Algorithm
___
    **Input** the collapsed quasi-skeleton of the rooted subtrees $T_i = (V_i, L_i, E_i)$ for $i = 1, ..., n_r$
    **Output** the collapsed quasi-skeleton $P$ of the latent polytree
 1: Initialize the set of clusters $\mathcal{P}$ with the hidden clusters of all $T_i$, i.e., $\mathcal{P} := \{\{C_1\}, \{C_2\}, ..., \{C_k\}\}$
 2: **while** there are two elements $C_i, C_j \in \mathcal{P}$ such that $|N(C_i) \cap N(C_j)| \geq 2$ **do**
 3:      remove $C_i, C_j$ from $\mathcal{P}$ and add $C_i \cup C_j$ to $\mathcal{P}$
 4:      define $N(C_i \cup C_j) := N(C_i) \cup N(C_j)$
 5: **end while**
 6: Define the polytree $P = (\cup_i V_i, \mathcal{P}, E)$ where $E := \{\{y_a, y_b\} \mid \exists\, i : y_a, y_b \in V_i, y_a - y_b \in E_i\} \cup \{\{y_a, C_b\} \mid \exists\, i, h : y_a \in V_i, y_h \in L_i, L_i \subseteq C_b, C_b \in \mathcal{P}, y_a - y_h \in E_i\}$
___

The following theorem guarantees that, for a minimal latent polytree, the output of HCMA is the collapsed quasi-skeleton of the polytree. See Supplementary Material for the proof of this theorem.

**Theorem 16.** *Let $\vec{P}_\ell = (V, L, \vec{E})$ be a minimal latent polytree and let $T_i = (V_i, L_i, E_i)$ for $i = 1, ..., n_r$ be the collapsed quasi-skeletons of the rooted subtrees of $\vec{P}_\ell$. Then HCMA outputs the collapsed quasi-skeleton of $\vec{P}_\ell$.*

### 3.4 Task 4: Determine the quasi-skeleton of the latent polytree from the collapsed quasi-skeleton of the latent polytree (recover type-I hidden nodes)

After performing the HCMA, the output is the collapsed quasi-skeleton of the latent polytree, thus, the structure of the hidden nodes within each hidden cluster is not known yet. Note that the restriction of the original polytree to the closure of a hidden cluster is a smaller polytree. The goal of this task is to recover the structure of the hidden clusters by focusing on each individual closure (i.e., recover Type-I hidden nodes and their connectivities). Given the closure of a hidden cluster, the basic strategy is to detect one root of the hidden cluster along with the visible nodes (if any) linked to this root. Then, we label such a root as a visible node, add edges between this node and its visible neighbors, and subsequently apply the same strategy recursively to the descendants of such a detected root.

Since we focus on the closure of a specific hidden cluster, say $C$, we define the following sets $\tilde{V}_r = V_r \cap N(C)$ for $r = 1, ..., n_r$ where $n_r$ is the number of rooted subtrees in the latent polytree and $V_r$ are the sets of visible nodes in each rooted subtree (obtained from Task 1). A fundamental result for detection of a root of a hidden cluster is the following theorem. See Supplementary Material for the proof of this theorem.

**Theorem 17.** *Let $\vec{P}_\ell$ be a minimal latent polytree and let $\vec{T}_r = (V_r, L_r, \vec{E}_r)$ with $r = 1, ..., n_r$ be all the rooted subtrees of $\vec{P}_\ell$. Let $C$ be a hidden cluster in the collapsed quasi-skeleton of $\vec{P}_\ell$. Define $\tilde{V}_r := V_r \cap N(C)$ for $r = 1, ..., n_r$ where $n_r$ is the number of roots in $\vec{P}_\ell$. Then, $T_r$ contains a hidden root of $C$ if and only if $\tilde{V}_r \neq \emptyset$ and for all $\tilde{V}_{r'}$ with $r' \neq r$ we have $|\tilde{V}_r \setminus \tilde{V}_{r'}| > 1$ or $|\tilde{V}_{r'} \setminus \tilde{V}_r| \leq 1$.*

To make the application of this theorem more clear, consider the latent polytree introduced in Figure 3 (True). After applying the first three tasks, we obtain the collapsed quasi-skeleton of the latent polytree as depicted in Figure 3 (Task 3). Observe that the rooted subtrees $\vec{T}_1$ (with root $y_1$) and $\vec{T}_2$ (with root $y_2$) satisfy the conditions of Theorem 17 indicating that they contain a root of the hidden cluster. The following lemma allows one to find the visible nodes linked to a hidden root in the closure of a hidden cluster. See Supplementary Material for the proof of this lemma.

**Lemma 18.** *Let $\vec{P}_\ell$ be a minimal latent polytree. Consider a hidden root $y_h$ of a hidden cluster $C$ in the collapsed quasi-skeleton of $\vec{P}_\ell$ where $y_h$ belongs to the rooted subtree $T_r = (V_r, L_r, \vec{E}_r)$. Define $\tilde{V}_{r'} := V_{r'} \cap N(C)$ for $r' = 1, ..., n_r$ where $n_r$ is the number of roots in $\vec{P}_\ell$. The visible nodes linked to $y_h$ are given by the set $W \setminus \overline{W}$ where*

$$I := \{r\} \cup \{r' \text{ such that } |\tilde{V}_r \setminus \tilde{V}_{r'}| = |\tilde{V}_{r'} \setminus \tilde{V}_r| = 1\}, \qquad W := \bigcup_{i \in I} \tilde{V}_i, \qquad \overline{W} := \bigcup_{i \notin I} \tilde{V}_i.$$

We follow the example of Figure 3 to show the steps of Task 4 in more details. Without loss of generality, choose $T_r = T_1$. Consider the closure of $C_{A'}$ obtained at the end of Task 3 and then apply Lemma 18 to obtain $I = \{1, 2\}$, $W = \{y_1, y_2, y_{10}, y_{12}, y_{13}, y_{14}, y_{15}, y_{16}, y_{17}\}$, $\overline{W} = \{y_5, y_6, y_9, y_{11}, y_{12}, y_{13}, y_{14}, y_{15}, y_{16}, y_{17}\}$, and thus $W \setminus \overline{W} = \{y_1, y_2, y_{10}\}$. Therefore, the visible nodes linked to the hidden root in $T_1$ are $y_1, y_2$ and $y_{10}$. Now we introduce the Hidden Cluster Learning Algorithm (HCLA), presented in Algorithm 2, to learn the structure of a hidden cluster.

Again, consider the closure of the hidden cluster $C_{A'}$ as depicted in Figure 4 (Task 4a) which we obtained at the end of Task 3. Then, apply Hidden Node Detection procedure to $C_{A'}$ and observe that the output at the end of Step 23 of Algorithm 2 is in Figure 4 (Task 4b). The output of the merging in Steps 24-27 is depicted in Figure 4 (Task 4c) and the output of the merging in Step 28 is depicted in Figure 4 (Task 4d). Now, we can apply the same procedure recursively to the remaining hidden clusters to obtain the final output of Task 4, the quasi-skeleton of the polytree, as depicted in Figure 3 (Task 4).

Here, we show that the output of HCLA is the quasi-skeleton of the latent polytree. See Supplementary Material for the proof of this theorem.

**Theorem 19.** *Let $\vec{P}_\ell = (V, L, \vec{E})$ be a minimal latent polytree. When HCLA is applied to all hidden clusters of the collapsed quasi-skeleton of $\vec{P}_\ell$, the output $P = (V, E)$ is the quasi-skeleton of $\vec{P}_\ell$. Furthermore, HCLA also outputs, for each pair $y_i, y_j \in V$, the relation $\mathcal{I}(y_i, \emptyset, y_j)$ if and only if the path connecting $y_i$ and $y_j$ in $\vec{P}_\ell$ contains an inverted fork.*

---

**Algorithm 2** Hidden Cluster Learning Algorithm

---

**Input** the collapsed quasi-skeleton of a minimal polytree $\vec{P}_\ell$, collapsed quasi-skeletons of the rooted subtrees $T_i = (V_i, L_i, E_i)$ for $i = 1, ..., n_r$, and the set of the hidden clusters $\mathcal{P} = \{C_1, ..., C_{n_C}\}$

**Output** $P$ and the independence relations of the form $\mathcal{I}(y_a, \emptyset, y_b)$ or $\neg\mathcal{I}(y_a, \emptyset, y_b)$ for all nodes $y_a, y_b \in \bigcup_i V_i$

1: **while** $\mathcal{P} \neq \emptyset$ **do**
2:     Call Hidden Node Detection Procedure($C_1$) where $C_1$ is the first element of $\mathcal{P}$
3: **end while**
4: **procedure** Hidden Node Detection($C$)
5:     Compute $\tilde{V}_i = V_i \cap N(C)$
6:     Find $\tilde{V}_r$ which satisfies $|\tilde{V}_r \setminus \tilde{V}_{r'}| > 1$ or $|\tilde{V}_{r'} \setminus \tilde{V}_r| \leq 1$ for all $r' \neq r$ (as in Theorem 17)
7:     Initialize $W := \tilde{V}_r$, $\overline{W} := \emptyset$, and $I := \{r\}$
8:     **for** all $i = 1, ..., n_r$ with $i \neq r$ **do**
9:         **if** $|\tilde{V}_r \setminus \tilde{V}_i| = 1$ and $|\tilde{V}_i \setminus \tilde{V}_r| = 1$ (as in Lemma 18) **then**
10:             $W := W \cup \tilde{V}_i$ and $I := I \cup \{i\}$
11:         **else**
12:             $\overline{W} := \overline{W} \cup \tilde{V}_i$
13:         **end if**
14:     **end for**
15:     A new hidden node $y_h$ is revealed
16:     Add $y_h$ to all the rooted trees $T_i$ with $i \in I$, namely $V_i := V_i \cup \{y_h\}$
17:     Add the independence relation $\neg\mathcal{I}(y_h, \emptyset, y)$ for all $y \in V_i$ with $i \in I$, and add the independence relation $\mathcal{I}(y_h, \emptyset, y)$ for all other nodes $y$
18:     Link all nodes in $W \setminus \overline{W}$ to $y_h$ in all $T_i$ with $i \in I$, namely $E_i := E_i \cup \left\{\{y_h, y\} \mid y \in W \setminus \overline{W}\right\}$
19:     **for** all $i \in I$ **do**
20:         create $n_k = |W \cap \overline{W}|$ new clusters: $C_1^{(i)}, ..., C_{n_k}^{(i)}$
21:         link $y_h$ to $C_1^{(i)}, ..., C_{n_k}^{(i)}$
22:         link each cluster $C_1^{(i)}, ..., C_{n_k}^{(i)}$ to a distinct element in $W \cap \overline{W}$
23:     **end for**
24:     **while** $\exists y_a, y_b \in N(C_j^{(i)}) \cup N(C_k^{(i)})$ such that $y_a, y_b \in \tilde{V}_m$ where $m \notin I$ **do**
25:         merge the two hidden clusters $C_j^{(i)}$ and $C_k^{(i)}$
26:         update the structure of $T_i$ with the new hidden clusters
27:     **end while**
28:     Let $P = (V, \mathcal{P}, E)$ be the output of HCMA applied to $T_i = (V_i, L_i, E_i)$, for $i = 1, ..., n_r$
29: **end procedure**

---

### 3.5 Task 5: Obtain the pattern of the latent polytree from the recovered quasi-skeleton of the latent polytree (recover type-II hidden nodes and edge orientations)

Once the quasi-skeleton of the latent polytree has been obtained, the only missing nodes to recover the full skeleton are the type-II hidden nodes of the original polytree. Interestingly, the detection of such hidden nodes can be performed concurrently with the recovery of the edge orientations. In particular, we apply Rebane and Pearl's algorithm in [13] to orient the edges of the quasi-skeleton of the polytree. Then, we have that the edges receiving double orientations imply the presence of a type-II hidden node between the two linked nodes. Thus, the Hidden Root Recovery Algorithm (HRRA), presented in Algorithm 3, is simply an implementation of Rebane and Pearl's algorithm (Steps 1-4), as depicted in Figure 4 (Task 5a), with the additional detection of type-II hidden nodes (Steps 5-10).

As a consequence, we have this final result stated in Theorem 20 to prove that HRRA outputs the pattern of the latent polytree. See Supplementary Material for the proof of this theorem.

**Theorem 20.** *Let $\vec{P}_\ell$ be a minimal latent polytree. When the input is the quasi-skeleton of $\vec{P}_\ell$ with the independence statements of the form $\mathcal{I}(y_i, \emptyset, y_j)$ or $\neg\mathcal{I}(y_i, \emptyset, y_j)$ for all the pairs of nodes $y_i$ and $y_j$, the output of HRRA is the pattern of $\vec{P}_\ell$.*

For a complete step by step example of this algorithm see the Supplementary Material.

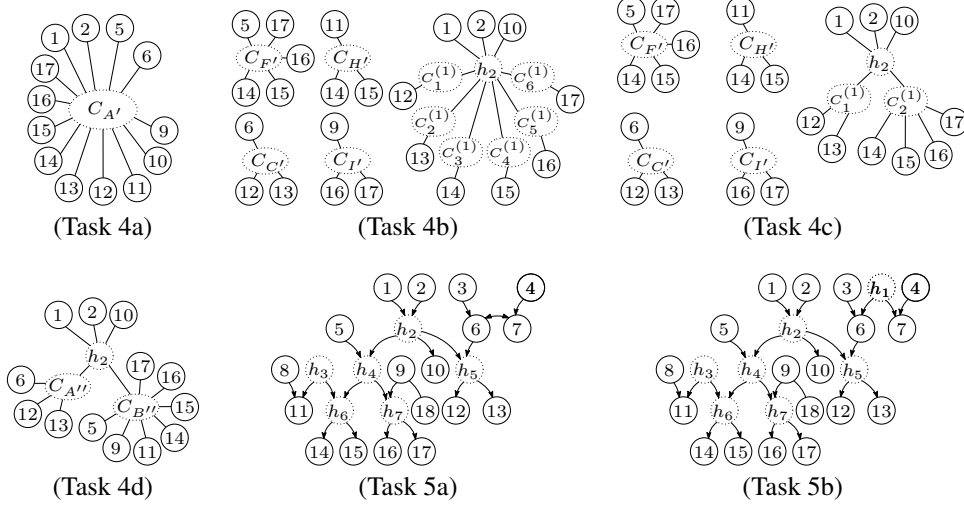

<div align="center">(Task 4a)        (Task 4b)        (Task 4c)</div>

<div align="center">(Task 4d)        (Task 5a)        (Task 5b)</div>

Figure 4: The closure of the hidden cluster $C_{A'}$ of the latent polytree in Figure 3 (True) obtained after Task 3 (Task 4a), the hidden clusters obtained after Step 23 of HCLA (Task 4b), merging of the overlapping hidden clusters as in Steps 24-27 of HCLA (Task 4c), merging of the overlapping hidden clusters as in Step 28 of HCLA (Task 4d), orienting the edges in the quasi-skeleton of the latent polytree as in Steps 1-4 of HRRA (Task 5a), and recovering type-II hidden nodes (Task 5b).

---

**Algorithm 3** Hidden Root Recovery Algorithm

---

    **Input** $P = (V, E)$, the quasi-skeleton of a latent polytree, and the independence relations of the form $\mathcal{I}(y_i, \emptyset, y_j)$ or $\neg\mathcal{I}(y_i, \emptyset, y_j)$ for all nodes $y_i, y_j \in V$

    **Output** the partially directed polytree $\bar{P} = (V, E, \vec{E})$

1: **while** additional edges are oriented **do**
2:     if $y_i - y_k, y_j - y_k \in E$ and $\mathcal{I}(y_i, \emptyset, y_j)$, then add $y_i \to y_k$ and $y_j \to y_k$ to $\vec{E}$
3:     if $y_i \to y_k \in \vec{E}$, $y_k - y_j \in E$ and $\neg\mathcal{I}(y_i, \emptyset, y_j)$, then add $y_k \to y_j$ to $\vec{E}$
4: **end while**
5: Remove the edges that are oriented in $\vec{E}$ from $E$
6: **for** all $y_i, y_j$ such that $y_i \to y_j, y_j \to y_i \in \vec{E}$ **do**
7:     a new hidden node of Type-II is detected which is a parent of $y_i$ and $y_j$
8:     remove $y_i \to y_j, y_j \to y_i$ from $\vec{E}$
9:     add a new node $y_h$ to $V$
10:     add $y_h \to y_j, y_h \to y_i$ to $\vec{E}$
11: **end for**

---

# 4 Conclusions and Discussion

We have provided an algorithm to reconstruct the pattern of a latent polytree graphical model. The algorithm only requires the second and third order statistics of the observed variables and no prior information about the number and location of the hidden nodes is assumed. An important property of the proposed approach is that the algorithm is sound under specific degree conditions on the hidden variables. If such degree conditions are not met, it is shown, in the Supplementary Material, that there exists another latent polytree with fewer number of hidden nodes entailing the same independence relations. In this sense, the proposed algorithm always recover a minimal graphical model in the sense of hidden nodes following a form of Occam's razor principle. Future work will study how this algorithm performs under limited amount of data and how to deal with situations when the measurements are not exact.

## Acknowledgments

This work has been partially supported by NSF (CNS CAREER #1553504).

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
