[Supplementary Material]

# An Algorithm to Learn Polytree Networks with Hidden Nodes - Supplementary Material

**Firoozeh Sepehr**
Department of EECS
University of Tennessee Knoxville
1520 Middle Dr, Knoxville, TN 37996
dawn@utk.edu

**Donatello Materassi**
Department of EECS
University of Tennessee Knoxville
1520 Middle Dr, Knoxville, TN 37996
dmateras@utk.edu

## 5.1  Task 1: Determine the visible nodes of each rooted subtree

### 5.1.1  Algorithm 4: Pairwise-Finite Distance Algorithm (PFDA)

We report a version of Pairwise-Finite Distance Algorithm (PFDA) that has been modified from the one presented in [16] just to match the notation of this article.

---
**Algorithm 4** Pairwise-Finite Distance Algorithm

> **Input** the ordered set of nodes $V = \{y_1, ..., y_n\}$ and the independence statements of the form $\mathcal{I}(y_i, \emptyset, y_j)$ or $\neg\mathcal{I}(y_i, \emptyset, y_j)$ for all $y_i, y_j \in V$
> **Output** the set of all non-eliminated lists $S_{i,j}$
1: **for** every node $y_i \in V$ such that $\forall y_j \in V \setminus \{y_i\}$ we have that $\mathcal{I}(y_i, \emptyset, y_j)$ **do**
2:      define $S_{i,0} := \{y_i\}$
3: **end for**
4: **for** each pair $y_i, y_j \in V$ with $i < j$, and $\neg\mathcal{I}(y_i, \emptyset, y_j)$ **do**
5:      Define $S_{i,j} := \{y_i, y_j\}$
6:      **for** each $y_k \in V \setminus S_{i,j}$ **do**
7:          if $\forall y \in S_{i,j} : \neg\mathcal{I}(y_k, \emptyset, y)$ , then add $y_k$ to $S_{i,j}$
8:      **end for**
9: **end for**
10: **for** each pair $y_i, y_j \in V$ with $i < j$ **do**
11:      if $S_{i,j} = S_{k,\ell}$ for some $k$ and $\ell$, then eliminate $S_{k,\ell}$
12: **end for**

---

### 5.1.2  Explanation and intuition behind PFDA

The main purpose of the PFDA in [16] is to recover the lists of all visible nodes in each rooted tree of a minimal latent polytree $(V, L, \vec{E})$ from independence relations of the form $\mathcal{I}(y_i, \emptyset, y_j)$ or $\neg\mathcal{I}(y_i, \emptyset, y_j)$ for all $y_i, y_j \in V$. In general, PFDA can deal with polyforests (a graph composed of multiple polytrees), but in this article we focus on its application to a single polytree.

Steps 1-3 deal with the special case when the polytree has exactly one node. Steps 4-9 are the main routine, where the algorithm initializes a list $S_{i,j}$ with two nodes $y_i, y_j \in V$ such that $\neg\mathcal{I}(y_i, \emptyset, y_j)$. Since $\neg\mathcal{I}(y_i, \emptyset, y_j)$ holds, both $y_i$ and $y_j$ belong to the same rooted subtree (potentially they belong to even more than one rooted subtree). The routine keeps adding nodes $y_k$ to $S_{i,j}$ so long as $y_k$ satisfies $\neg\mathcal{I}(y_k, \emptyset, y)$ for all $y$ currently in $S_{i,j}$. Such a routine is repeated for all pairs of distinct $y_i, y_j$ computing all the corresponding lists $S_{i,j}$. Steps 10-12 simply remove potential duplicate lists $S_{i,j}$.

In [16], the original version of PFDA does not take as an input the relations $\mathcal{I}(y_i, \emptyset, y_j)$ or $\neg\mathcal{I}(y_i, \emptyset, y_j)$. Instead, it takes as input a metric $d$ with the property that $d(y_i, y_j) < \infty$ if and only if $y_i, y_j$ are in the

same rooted subtree. Since $\neg\mathcal{I}(y_i, \emptyset, y_j)$ holds if and only if $y_i, y_j$ are in the same rooted subtree, it is immediate to verify that the relations $\mathcal{I}(y_i, \emptyset, y_j)$ or $\neg\mathcal{I}(y_i, \emptyset, y_j)$ can replace the role of the metric in the original algorithm, that is precisely how we have modified it in this article.

### 5.1.3 Proof of Theorem 13

*Proof.* In [16], the authors show that PFDA outputs the lists of visible nodes belonging to each rooted subtree of the latent polytree $\vec{P}_\ell = (V, L, \vec{E})$ when the distances between pairs of nodes, i.e., $d(y_i, y_j)$ for all $y_i, y_j \in V$ are given by a metric $d$ which satisfies the property that $d(y_i, y_j) < \infty$ if and only if $y_i$ and $y_j$ are in the same rooted subtree. Define $d(y_i, y_j) = 0$ if and only if $\neg\mathcal{I}(y_i, \emptyset, y_j)$ holds and define $d(y_i, y_j) = \infty$ if and only if $\mathcal{I}(y_i, \emptyset, y_j)$ holds. Using this metric, the original PFDA in [16] becomes PFDA of this article with all the related guarantees. □

## 5.2 Task 2: Determine the collapsed quasi-skeleton of each rooted subtree

### 5.2.1 Algorithm 5: Reconstruction Algorithm for Latent Rooted Trees

We report a version of Reconstruction Algorithm for Latent Rooted Trees that has been modified from the one presented in [17] just to match the notation of this article.

---

**Algorithm 5** Reconstruction Algorithm for Latent Rooted Trees

---
    **Input** the set of visible nodes in a rooted subtree $V_r$ and the independence statements of the form $\mathcal{I}(y_i, y_k, y_j)$ or $\neg\mathcal{I}(y_i, y_k, y_j)$ for $y_i, y_j, y_k \in V_r$
    **Output** $(V_r, L_r, E_r)$ the collapsed quasi-skeleton of $T_r$
1: Initialize $V_{temp} := V_r$, $L_r := \{\}$, and $E_r := \{\}$
2: If $|V_{temp}| = 2$, i.e., $V_{temp} = \{y_i, y_j\}$, then add the edge $y_i - y_j$ to $E_r$ and if $n \leq 2$, stop and output the results
3: Determine a visible terminal node $y_k$ in the rooted tree by verifying the condition $\neg\mathcal{I}(y_i, y_k, y_j)$ for all $y_i, y_j \in V_{temp} \setminus \{y_k\}$
4: Search for a visible node $y_\ell \in V_{temp} \setminus \{y_k\}$ linked to $y_k$ by verifying the condition $\mathcal{I}(y_k, y_\ell, y_j)$ for all $\forall y_j \in V_{temp} \setminus \{y_k, y_\ell\}$
5: **if** $y_\ell$ exists **then**
6:     add the link $y_\ell - y_k$ to $E_r$, remove $y_k$ from $V_{temp}$, and go to Step 2.
7: **else**
8:     create a new hidden node $y_h$ in $L_r$ and add the link $y_k - y_h$ to $E_r$
9:     compute the set $K \subseteq V_{temp}$ such that $y_j \in K$ implies that $\neg\mathcal{I}(y_j, y_i, y_k)$ for all $y_i \neq y_j, y_k$ where $y_i \in V_{temp} \setminus K$
10:     add $y_h - y_j$ to $E_r$ for $y_j \in K$
11:     set $(V^{(j)}, L^{(j)}, E^{(j)})$ as the output of this algorithm applied to each $V^{(j)}$ defined as the union of $\{y_j\}$ and the set of nodes in $V_{temp}$ separated from $y_k$ by $y_j \in K$
12:     set $E_r := \bigcup_{y_j \in K} E^{(j)} \cup E_r$ and $L_r := \bigcup_{y_j \in K} L^{(j)} \cup L_r$
13: **end if**

---

### 5.2.2 Explanation and intuition behind the algorithm

The main goal of the Reconstruction Algorithm for Latent Rooted Trees developed in [17] is to reconstruct the collapsed quasi-skeleton of a latent rooted tree from independence relation of the form $\mathcal{I}(y_i, y_k, y_j)$ or $\neg\mathcal{I}(y_i, y_k, y_j)$ for $y_i, y_j, y_k \in V_r$ where $V_r$ is the set of visible nodes of the rooted tree. The algorithm and its properties are described in detail in [17]. Here we just provide a brief description of the intuition behind it.

In particular, one fundamental result in [17] is that the Reconstruction Algorithm for Latent Rooted Trees can reconstruct the collapsed skeleton of every rooted tree so long as each hidden cluster has degree greater than or equal to 3. All other hidden clusters are undetected: for each hidden cluster with degree equal to 2, the two nodes linked to such a cluster are linked together by the algorithm; for each cluster with degree equal to 1, the algorithm ignores the cluster. In the context of this dissertation, all hidden clusters in each rooted subtree of a minimal latent polytree have degree greater than or equal to 3 with the exception of the special case where we have a hidden root with two

visible children. This is basically the main reason why we have introduced quasi-skeletons: in a quasi-skeleton this special case is removed. The following lemma makes sure that hidden clusters in quasi-skeletons of rooted subtrees have degree at least equal to 3 when considering minimal latent polytrees.

**Lemma 21.** *Let $\vec{P}_\ell = (V, L, \vec{E})$ be a latent polytree and let $T_r = (V_r, L_r, \vec{E}_r)$ be a rooted subtree of $\vec{P}_\ell$ with the root $y_r$. If $\vec{P}_\ell$ is minimal, then each hidden cluster in the quasi-skeleton of $T_r$ has degree at least* 3.

*Proof.* Since $\vec{P}_\ell$ is minimal, we distinguish the following two cases:

1. if the hidden node $y_h$ has $\deg^+_{\vec{P}_\ell}(y_h) \geq 2$ and $\deg_{\vec{P}_\ell}(y_h) \geq 3$ then it is trivially true that the hidden cluster that $y_h$ belongs to has degree at least 3 in the quasi-skeleton of the rooted subtree $T_r$.

2. if the hidden node $y_h$ has $\deg^+_{\vec{P}_\ell}(y_h) = 2$ and $\deg^-_{\vec{P}_\ell}(y_h) = 0$, then we have two subscenarios:

   a. if the two children of $y_h$ are visible, then the hidden cluster containing $y_h$ is made of only $y_h$. However, $y_h$ is a Type-II hidden node and therefore such a cluster does not appear in the quasi-skeleton of $T_r$.

   b. if at least one child of $y_h$ is hidden, say $y_c$, then the hidden cluster containing $y_h$ is the same hidden cluster that contains $y_c$. Since $y_c$ is a Type-I hidden node, we fall back to case 1, proving that the hidden cluster containing $y_h$ has degree at least 3.

$\square$

Thus, Lemma 21 allows us to apply Reconstruction Algorithm for Latent Rooted Trees to recover the quasi-skeletons of the rooted subtrees.

Now we provide a brief description of the intuition behind the Reconstruction Algorithm for Latent Rooted Trees. Step 1 is just the initialization and Step 2 is a basic induction step solving the problem when the minimal rooted tree has 1 or 2 visible nodes. Observe that, in the collapsed skeleton of a latent rooted tree where all hidden clusters have degree at least 3, all nodes with degree equal to 1 (namely, terminal nodes) are visible and they are either linked to another visible node or linked to a hidden cluster which is connected to at least 2 other visible nodes. Step 3 searches for a terminal visible node $y_k$: as proven in [17], a node $y_k$ is terminal in a rooted tree where hidden clusters have degree at least 3 if and only if there is no pair of visible nodes $y_i, y_j \in V \setminus \{y_k\}$ such that $\neg \mathcal{I}(y_i, y_k, y_j)$. This is precisely what is tested in this step. For example, considering the polytree in Figure 3 (True), one of the lists of visible nodes is the set $V_r = \{y_9, y_{16}, y_{17}, y_{18}\}$ associated with the rooted tree with the root $y_9$. The quasi-skeleton of this rooted tree is depicted in Figure 5 (a) and the list containing its visible nodes obtained at the end of Task 1 is depicted in Figure 5 (b). Observe that node $y_{18}$ satisfies the conditions of Step 3 since it cannot $d$-separate any pair of other nodes in $V_r$ (in other words, the node $y_{18}$ cannot make any pair of visible nodes independent). Thus, node $y_{18}$ is terminal in this rooted subtree.

Figure 5: Quasi-skeleton of the actual rooted tree with root in node $y_9$ (a), the list of visible nodes belonging to the rooted tree with root in node $y_9$ (b), node $y_{18}$ satisfies the conditions of being a terminal node and node $y_9$ satisfies the conditions of being the visible node linked to it (c), and node $y_{17}$ is found to be terminal but not linked to a visible node, thus, a hidden node linked to $y_{17}$ is detected (d).

Once a visible node $y_k$ with $\deg(y_k) = 1$ is found, Step 4 looks for a single visible node $y_\ell$ linked to $y_k$. We have that $y_\ell$ exists if and only if $\forall y_j \in V_r \setminus \{y_k, y_\ell\} : \mathcal{I}(y_k, y_\ell, y_j)$. This is the case for node $y_{18}$, since we have that $y_9$ makes $y_{18}$ independent of all the other nodes in $V_r$. If $y_\ell \in V$ exists, then the

test at Step 4 finds it and then at Step 6 the edge $\{y_k, y_\ell\}$ is added to $E_r$, and the algorithm is run again on $V_r \setminus \{y_k\}$. In our example, the nodes $y_9$ and $y_{18}$ are linked together and the algorithm is applied to $V_r \setminus \{y_{18}\}$, as depicted in Figure 5 (c).

When the algorithm runs again on $V_r \setminus \{y_{18}\}$, it is found that, for example, node $y_{17}$ is terminal. However, Step 4 cannot find any visible node linked to $y_{17}$. Thus, $y_{17}$ must be connected to a hidden cluster: Step 8 is where a new hidden cluster is created. Step 9 finds the set $K$ which contains all other visible neighbors of this hidden cluster. In our example, we have that $K = \{y_9, y_{16}\}$, as depicted in Figure 5 (d). At Step 10, node $y_h$ is linked to all $y_j \in K$, as depicted in Figure 5 (d), and the algorithm is applied recursively to $V_r^{(j)} := \{y_i \mid \mathcal{I}(y_k, y_j, y_i)\}$ for all $y_j \in K$ at Step 11. Step 12 sets the output to the union of all the outputs obtained at Step 11.

## 5.3   Proof of Theorem 14

In order to formally prove Theorem 14, first we need to introduce two additional results: Theorems 22 and 23. In Theorem 22 a criterion is provided to determine if the unique node linked to a visible terminal node is also visible. This criterion uses only the independence statements involving the visible nodes.

**Theorem 22.** *Let $T$ be the quasi-skeleton of a rooted subtree of a minimal latent polytree. Let the visible nodes in $T$ be $V$. Let $y_j$ be a terminal node and let $y_k$ be the unique node linked to $y_j$. The node $y_k$ is visible if and only if there exists a visible node $y_{k'}$ such that $\mathcal{I}(y_j, y_{k'}, y_i)$ for all $y_i \in V \setminus \{y_j, y_{k'}\}$. Furthermore we have that $y_k = y_{k'}$.*

*Proof.* $\Rightarrow$: If $y_k$ is visible, then set $y_{k'} = y_k$ and we have that $\mathcal{I}(y_j, y_{k'}, y_i)$ for all $y_i \in V \setminus \{y_j, y_{k'}\}$.

$\Leftarrow$: Let $y_{k'}$ be a visible node such that $\mathcal{I}(y_j, y_{k'}, y_i)$ for all $y_i \in V \setminus \{y_j, y_{k'}\}$. By contradiction, assume that $y_k$ is not visible. Thus, $y_k$ belongs to a hidden cluster and the node $y_j$ is directly linked to such a cluster. Therefore, there are at least two other visible nodes directly linked to this cluster. Let $y_i \neq y_{k'}$ be one of these visible nodes. The path from $y_i$ to $y_j$ involves only hidden nodes, thus it is not true that $\mathcal{I}(y_j, y_{k'}, y_i)$ which is a contradiction. So far, we have shown that the existence of a visible $y_{k'}$ such that $\mathcal{I}(y_j, y_{k'}, y_i)$ for all $y_i \in V \setminus \{y_j, y_{k'}\}$ implies that $y_k$ is visible. We also need to show that $y_{k'} = y_k$. Again, by contradiction, assume that $y_{k'} \neq y_k$. Then, it does not hold that $\mathcal{I}(y_j, y_{k'}, y_k)$ which is a contradiction. □

The following theorem complements Theorem 22 by stating that if there exists a set $K$ of visible nodes that can not be separated from $y_j$ by any other visible nodes and $K$ has at least two elements, then all the nodes in $K$ and $y_j$ are linked to the same hidden cluster.

**Theorem 23.** *Let $T$ be the quasi-skeleton of a rooted subtree of a minimal latent polytree. Let the visible nodes in $T$ be $V$. Also, let $y_j$ be a terminal node linked to a hidden cluster $C$ and let $K$ be the set of visible nodes connected to $C$ excluding $y_j$. Then, it holds that*

$$K = \{y_k \in V \setminus \{y_j\} \mid \forall y_i \in V \setminus \{y_k, y_j\} : \neg\mathcal{I}(y_j, y_i, y_k)\} \tag{1}$$

*such that for all $y_i \in V \setminus \{y_k, y_j\}$ there exists $y_k \in K$ such that $\mathcal{I}(y_j, y_k, y_i)$.*

*Proof.* We first show that if $y_k \neq y_j$ is a visible node connected to the hidden cluster $C$, then $\neg\mathcal{I}(y_j, y_i, y_k)$ for all $y_i \in V \setminus \{y_k, y_j\}$. By contradiction assume that there is $y_i \in V$ such that $\mathcal{I}(y_j, y_i, y_k)$. Now, there is $y_{k'} \in C$ and $y_{j'} \in C$ such that $y_k - y_{k'}$ and $y_j - y_{j'}$ belong to the set of edges $E$. Since both $y_{j'}$ and $y_{k'}$ belong to cluster $C$, there is a path from $y_{j'}$ to $y_{k'}$ that does not involve any visible nodes. Thus, there is no $y_i \in V$ such that $\mathcal{I}(y_j, y_i, y_k)$.

Now we show that if $\neg\mathcal{I}(y_j, y_i, y_k)$ for all $y_i \in V \setminus \{y_j, y_k\}$, then $y_k$ is connected to cluster $C$. By contradiction, assume that it is not. Consider the path from $y_j$ to $y_k$. Since $y_j$ is a terminal node and it is connected to $C$, there exists $y_{j'} \in C$ such that $y_j - y_{j'}$ is an edge of $E$ and this is the only edge in the graph involving $y_j$. Thus, the path from $y_j$ to $y_k$ contains $y_{j'}$. Consider the path from $y_{j'}$ to $y_k$ and let $y_{k'}$ be the first visible node on this path. The node $y_{k'}$ is directly linked to cluster $C$ and if $y_k \neq y_{k'}$, then we have that $\mathcal{I}(y_j, y_{k'}, y_k)$. Thus, we necessarily have that $y_k = y_{k'}$. □

Now, we can provide the proof of Theorem 14.

*Proof.* The proof goes by induction on the number of nodes denoted by $n$. For $n \leq 2$ the algorithm trivially outputs the correct result. For $n > 2$, we combine Proposition 2.1 and Theorem 4.1 in [17], to guarantee that a visible terminal node $y_k$ is always found at Step 3. Theorem 22 provides a sufficient and necessary condition to find a visible $y_\ell$ directly linked to $y_k$. If such a visible node $y_\ell$ exists, then the edge $y_\ell - y_k$ belongs to the skeleton of the original graph. Then the theorem is applied recursively to a network with $(n-1)$ nodes which is obtained by removing the terminal node $y_k$ from the original one. If such a $y_\ell$ does not exist, then $y_k$ is necessarily connected to a hidden cluster. Theorem 23 provides a necessary and sufficient condition to find the set $K$ of all visible nodes, other than $y_k$, linked to such a hidden cluster. A new hidden node $y_h$ is introduced to the set $L$ in order to represent the hidden cluster in the collapsed rooted subtree and the link $y_k - y_h$ is added to the set of edges $E$. Also the edges $y_h - y_j$ for all $y_j \in K$ are added to $E$. The algorithm is then applied recursively to each set $V^{(j)}$ given by all visible nodes $y_i$ such that $\mathcal{I}(y_i, y_j, y_k)$. Observe that each $V^{(j)}$ contains fewer number of nodes than $n$, guaranteeing that the algorithm always terminates. $\qquad\square$

### 5.4 Task 3: Merge the overlapping hidden clusters of the collapsed rooted trees

#### 5.4.1 Proof of Theorem 15

First we prove a lemma stating that, in a minimal latent polytree, two hidden clusters in two rooted subtrees share at least one node if and only if they have at least two common neighbors.

**Lemma 24.** *Let $C_1$ and $C_2$ be two distinct hidden clusters in two rooted subtrees in a minimal latent polytree $\vec{P}_\ell$. The two clusters overlap, i.e., $|C_1 \cap C_2| \geq 1$, if and only if there exist two distinct nodes $y_1, y_2 \in N(C_1) \cap N(C_2)$.*

*Proof.* $\Rightarrow$: This implication is trivially verified because of the minimality conditions of the latent polytree. If the hidden node in common has two visible descendants $y_1$ and $y_2$, then the implication is immediate. If it has, instead, at least one hidden child which belongs to $C_1 \cap C_2$, then such a hidden child either has two visible descendants giving the implication or, again, a hidden child which belongs to $C_1 \cap C_2$. Repeating the argument, we eventually find the common nodes $y_1, y_2$.

$\Leftarrow$: By contradiction, assume that $C_1$ and $C_2$ do not overlap. Since $C_1$ and $C_2$ share no common node but have two common neighbors, there must be a loop in the latent polytree $\vec{P}_\ell$, contradicting the fact that it is a polytree. $\qquad\square$

Now we can provide the proof of Theorem 15.

*Proof.* From Lemma 24, the proof of Theorem 15 is straightforward. If there are two common neighbors, then the two hidden clusters in the two rooted subtrees overlap, thus they belong to the same hidden cluster in the latent polytree. $\qquad\square$

#### 5.4.2 Proof of Theorem 16

*Proof.* The algorithm HCMA proceeds by sequentially merging clusters of the collapsed quasi-skeletons of the rooted subtrees of $\vec{P}_\ell$ if they share at least 2 neighbors (Steps 2-5). According to Lemma 24, this is equivalent to merging these clusters when they overlap (i.e., they have at least one hidden node in common). Thus, the initial set $\mathcal{P}$ contains all the hidden clusters in all the quasi-skeletons of the rooted subtrees of $\vec{P}_\ell$. If two hidden clusters in the quasi-skeletons of two rooted subtrees overlap, then they are necessarily in the same cluster of the quasi-skeleton of the original polytree $\vec{P}_\ell$. Thus, we just need to show that HCMA groups together all the hidden clusters in quasi-skeletons of the rooted subtrees which are in the same hidden cluster in the quasi-skeleton of $\vec{P}_\ell$.

By contradiction, assume that this is not true. Then the output of HCMA contains a union of clusters that does not exist in the collapsed quasi-skeleton of $\vec{P}_\ell$. Let this union of clusters be $U$. Thus, there exists at least one hidden node $y_h$ in one hidden cluster $C$ of the quasi-skeleton of $\vec{P}_\ell$ that does not belong to $U$. Consider the path from $y_h$ to any node in $U$. By definition such a path consists of all hidden nodes. Let $y_a$ be the last node on such a path that does not belong to $U$ and $y_b$ be the node following $y_a$ on this path. We necessarily have that $y_a \rightarrow y_b$, otherwise $y_a$ would be a descendant of

$y_b$ and hence in $U$. Consider a rooted tree containing $y_a$. Such a rooted tree has a hidden cluster $C'$ which contains $y_b$ as well and consequently overlaps with $U$, but $C'$ has not been included in $U$ by HCMA. This is a contradiction, because if two clusters overlap, then they are grouped together by HCMA. Therefore, this proves the assertion. □

## 5.5 Task 4: Determine the quasi-skeleton of the latent polytree from the collapsed quasi-skeleton of the latent polytree (recover type-I hidden nodes)

### 5.5.1 Proof of Theorem 17

We first provide the following straightforward lemma.

**Lemma 25.** *Every hidden node in a minimal latent polytree $\vec{P}_\ell = (V, L, \vec{E})$ has at least two visible descendants.*

*Proof.* Since the latent polytree is minimal, for every hidden node $y_h \in L$ we have that $|\text{ch}_{\vec{P}_\ell}(y_h)| \geq 2$. Now, we distinguish the following two cases.

1. If $|\text{ch}_{\vec{P}_\ell}(y_h) \cap V| \geq 2$, then the statement is trivially true.

2. If $|\text{ch}_{\vec{P}_\ell}(y_h) \cap V| < 2$, then the statement is trivially true by iterating the same argument on one element of the set $\text{ch}_{\vec{P}_\ell}(y_h) \cap L$.

□

Now we leverage the result of Lemma 25 to prove Theorem 17.

*Proof.* $\Rightarrow$: Let $y_{h_r} \in L_r$ be a hidden root of $C$. Now, we distinguish the following two cases.

- If the root of $\vec{T}_r$, namely $y_r$, is visible, then $y_r$ is necessarily a parent of $y_{h_r}$. If the root of $\vec{T}_{r'}$, namely $y_{r'}$, is also a parent of $y_{h_r}$, then we have $|\tilde{V}_r \setminus \tilde{V}_{r'}| = 1$ and $|\tilde{V}_{r'} \setminus \tilde{V}_r| = 1$ because $\text{de}_C(y_r) \setminus \{y_r\} = \text{de}_C(y_{r'}) \setminus \{y_{r'}\}$. If, instead, the root of $\vec{T}_{r'}$, namely $y_{r'}$, is not a parent of $y_{h_r}$, then there exists a path connecting one of the children of $y_{h_r}$, namely $y_{c_1}$, to $y_{r'}$ and this path necessarily contains an inverted fork. Now, consider another child of $y_{h_r}$, namely $y_{c_2}$. Observe that this child exists since all the hidden nodes in the collapsed quasi-skeleton of the rooted subtrees of $\vec{P}_\ell$ have at least outdegree two (see Definitions 8 and 10). The child node $y_{c_2}$ is either visible itself or has at least two visible descendants according to Lemma 25. If $y_{c_2}$ is visible, then let $A := \{y_{c_2}, y_r\}$. If $y_{c_2}$ is not visible, then let $A := \{\text{de}_C(y_{c_2}) \cap \tilde{V}_r\}$. In either case, we have that $|\tilde{V}_r \setminus \tilde{V}_{r'}| \geq 2$ since $A \cap \text{de}_C(y_{r'}) = \emptyset$ because there exists an inverted fork on the path from $y_{h_r}$ to $y_{r'}$.

- If the root of $\vec{T}_r$, namely $y_r$, is hidden, then $|\text{ch}_C(y_r)| \geq 2$. If $|\text{ch}_C(y_r)| \geq 3$, let $y_{c_1}$ be the child of $y_r$ such that it is on the path from $y_r$ to $y_{r'}$. Observe that this path contains at least one inverted fork. Thus, if $y_{c_2}$ and $y_{c_3}$ are visible, then we have that $|V_r \setminus V_{r'}| \geq 2$. If, instead any of $y_{c_2}$ or $y_{c_3}$ are hidden, then they should have at least two visible descendants according to Lemma 25 which also results in having $|\tilde{V}_r \setminus \tilde{V}_{r'}| \geq 2$. On the other hand, if $|\text{ch}_C(y_r)| = 2$, then both of these children are hidden since we are working with the collapsed quasi-skeleton of $\vec{P}_\ell$. In this case, each of these hidden children have at least two visible descendants according to Lemma 25. Therefore, we have $|\tilde{V}_r \setminus \tilde{V}_{r'}| \geq 2$.

$\Leftarrow$: We prove, instead, that if $\vec{T}_r$ does not contain a hidden root of $C$, then $\exists \tilde{V}_{r'} : |\tilde{V}_r \setminus \tilde{V}_{r'}| \leq 1$ and $|\tilde{V}_{r'} \setminus \tilde{V}_r| > 1$. We distinguish the following two cases.

- If the root of $\vec{T}_r$, namely $y_r$, is visible, then we know that $y_r$ has exactly one hidden child, namely $y_{h_c}$, because $\vec{T}_r$ belongs to $N(C)$. Since this node is not a hidden root of $C$, then it has at least one hidden parent, namely $y_{h_p}$. Let $\vec{T}_{r'}$ be the rooted tree such that $y_{h_p} \in L_{r'}$. In this case, we know that $\tilde{V}_r = \{y_r \cup \text{de}_C(y_{h_c})\}$, $\text{de}_C(y_{h_c}) \subset \tilde{V}_{r'}$, and $y_r \notin \tilde{V}_{r'}$. Thus, we have that $|\tilde{V}_r \setminus \tilde{V}_{r'}| = |\{y_r\}| = 1$.

Furthermore, if $y_{h_p}$ is the root of $\vec{T}_{r'}$, then this case is similar to the second case of the first part of this proof where there exists a path from $y_r$ to $y_{h_p}$ which contains at least one inverted fork. In this case, we have that $|\tilde{V}_{r'} \setminus \tilde{V}_r| > 1$. If, instead, $y_{h_p}$ is not the root of $\vec{T}_{r'}$, then $y_{h_p}$ has at least two children because of minimality conditions and at least one parent. The other child of $y_{h_p}$ (i.e., not the node $y_{h_c}$) and one of the parents of $y_{h_p}$ are either visible or hidden that satisfy minimality conditions. In either case, we have that $|\tilde{V}_{r'} \setminus \tilde{V}_r| > 1$.

- If the root of $\vec{T}_r$ is hidden, then this would be a contradiction to the hypothesis since $\vec{T}_r$ does not contain a hidden root of $C$.

$\square$

### 5.5.2 Proof of Lemma 18

First, we introduce a lemma which provides the conditions for finding the parents of a hidden root of a hidden cluster in a minimal latent polytree.

**Lemma 26.** *Let $\vec{P}_\ell$ be a minimal latent polytree and define the rooted subtrees $\vec{T}_i$, the sets $\tilde{V}_i$ for $i = 1, ..., n_r$, the hidden root $y_h$ and the hidden cluster $C$ as in Lemma 18. Let $\tilde{V}_r$ contain $y_h$ which is a hidden root of the hidden cluster $C$. We have that $\tilde{V}_r \setminus \tilde{V}_{r'} = \{y_v\}$ and $\tilde{V}_{r'} \setminus \tilde{V}_r = \{y_{v'}\}$ if and only if $y_v$ and $y_{v'}$ are parents of $y_h$.*

*Proof.* $\Rightarrow$: We show that $y_v$ and $y_{v'}$ are the parents of the root of the hidden cluster $C$. Let $y_r$ and $y_{r'}$ be the roots of the restriction of $\vec{T}_r$ and $\vec{T}_{r'}$ to the closure of $C$, respectively. Consider the path from $y_r$ to $y_{r'}$. This path needs to have a length of at least 2, otherwise either $y_r$ or $y_{r'}$ would be a child of the other contradicting the fact that they are roots in the restriction of $\vec{P}_\ell$ to the closure of $C$. If the length of this path is greater than 2, then it needs to have the form $y_r \rightarrow y_{h_1} \cdots y_{h_2} \leftarrow y_{r'}$ where $y_{h_1}$ and $y_{h_2}$ are two distinct hidden nodes. As a result, either $y_{h_1}$ or $y_{h_2}$ is not a descendant of the other. Furthermore, because of the minimality conditions, in the closure of $C$, either there exist two visible descendants of $y_{h_1}$ that are not descendants of $y_{r'}$ or there exist two visible descendants of $y_{h_2}$ that are not descendants of $y_r$. This contradicts the fact that $|\tilde{V}_r \setminus \tilde{V}_{r'}| = |\tilde{V}_{r'} \setminus \tilde{V}_r| = 1$. Thus, the path between $y_r$ and $y_{r'}$ necessarily has length 2 and has the form $y_r \rightarrow y_{h_1} \leftarrow y_{r'}$ for some hidden node $y_{h_1}$. As a consequence, $y_r = y_v$, $y_{r'} = y_{v'}$ and also $y_{h_1} = y_h$ is the root of the hidden cluster $C$.

$\Leftarrow$: This implication is trivial. $\square$

Now we can provide the proof of Lemma 18.

*Proof.* For a fixed $\tilde{V}_r$, the set of indices $I \subseteq \{1, 2, ..., n_r\}$ with $n_r$ equal to the number of rooted subtrees is defined as the set $\{r\} \cup \{r'$ such that $|\tilde{V}_r \setminus \tilde{V}_{r'}| = |\tilde{V}_{r'} \setminus \tilde{V}_r| = 1\}$. It is trivial to show that if $|\tilde{V}_r \setminus \tilde{V}_{r'}| = |\tilde{V}_{r'} \setminus \tilde{V}_r|$, then the two sets $\tilde{V}_r$ and $\tilde{V}_{r'}$ can be written as

$$\tilde{V}_r = \{y_v\} \cup \underline{\tilde{V}}, \qquad \tilde{V}_{r'} = \{y_{v'}\} \cup \underline{\tilde{V}}. \tag{2}$$

In other words, there is exactly one element $y_v$ in $\tilde{V}_r$ which is not in $\tilde{V}_{r'}$. Similarly there exists exactly one $y_{v'}$ in $\tilde{V}_{r'}$ which is not in $\tilde{V}_r$.

Now, we show that $W \setminus \overline{W}$ is the set of all nodes linked to $y_h$ which is a root of the hidden cluster $C$. If a visible node $y_w$ is linked to $y_h$, it is either its parent or its child. If $y_w$ is a parent of $y_h$, then it is contained in $W$ and cannot be in $\overline{W}$ because of Lemma 26. If $y_w$ is a child of $y_h$, then it cannot be in $\overline{W}$ because $I$ contains no index associated with subtrees containing any of the parents of $y_h$ (see Lemma 26).

Now, we show, instead, that if $y_w \in W \setminus \overline{W}$, then $y_w$ is linked to $y_h$. Equivalently, we show that if $y_w$ is not linked to $y_h$, then $y_w \notin W \setminus \overline{W}$. Consider the following two cases.

- Node $y_w$ is a root of the closure of $C$. Since $y_w$ is not linked to $y_h$, it is not a parent of $y_h$ and from Lemma 26 we have that $y_w \in \overline{W}$.

- Node $y_w$ is not a root of the closure of $C$. Consider the path from $y_h$ to $y_w$ which has the form $y_h \to \cdots y_{h_1} \to y_w$. Since the node $y_{h_1}$ is hidden, the minimality conditions imply that $y_{h_1}$ has at least another parent, namely $y_p$ (hidden or visible). Since $y_p$ is not a parent of $y_h$, every rooted subtree $\vec{T}_i$ containing $y_p$ is such that $i \notin I$ (see Lemma 26). Thus, all the visible descendants of $y_p$ (including $y_w$) are necessarily in $\overline{W}$.

$\square$

### 5.5.3  Proof of Theorem 19

We first provide the following lemma to ensure that the Steps 24-27 of HCLA correctly merge the fictitious hidden clusters.

**Lemma 27.** *There exists two distinct nodes $y_a, y_b$ in $W \cap \overline{W}$ such that $y_a, y_b \in \tilde{V}_m$ where $m \notin I$, if and only if $y_a$ and $y_b$ are connected to the same hidden cluster.*

*Proof.* Observe that $N(C_j^{(i)}) \cup N(C_k^{(i)}) \subseteq W \cap \overline{W}$. Consider two distinct elements $y_a, y_b$ in $W \cap \overline{W}$.

$\Rightarrow$: If we have that $y_a, y_b \in \tilde{V}_m$ where $m \notin I$, then we know that $y_a$ and $y_b$ belong to a commom rooted subtree that does not contain the newly recovered hidden node $y_h$. Now, by contradiction, assume that the nodes $y_a$ and $y_b$ are not connected to the same hidden cluster. This implies that there exists a path connecting $y_a$ to $y_b$ through $y_h$. On the other hand, since $y_a$ and $y_b$ belong to a common rooted subtree that does not contain $y_h$, there exists another path connecting $y_a$ and $y_b$ which does not include $y_h$. This is a contradiction with the fact that any two nodes in a polytree are connected to each other via at most one path. Therefore, $y_a$ and $y_b$ are connected to the same hidden cluster.

$\Leftarrow$: Let $C$ be the hidden cluster to which both $y_a$ and $y_b$ are connected. Thus, we know that $C$ has $y_h$ as its parent, more specifically, $y_h$ is a parent of one hidden node in $C$. Let this hidden node be $y_{h_1}$. Since $y_{h_1}$ has the hidden node $y_h$ as its parent, it is required, by the minimality conditions, that $y_{h_1}$ has at least one other parent. This implies that $y_a$ and $y_b$ are contained in at least one rooted subtree $\vec{T}_m$ which does not contain $y_h$, namely, $m \notin I$ and also $y_a$ and $y_b$ are contained in $W \cap \overline{W}$. $\square$

Now we can provide the proof of Theorem 19.

*Proof.* HCLA calls the subroutine Hidden Node Detection on all hidden clusters until no more hidden nodes are discovered (Steps 1-3). The goal of Hidden Node Detection is to locate a hidden root $y_h$ in the collapsed quasi-skeleton $(V, L, E)$ of a polytree, determine the visible nodes linked to it and compute the new collapsed quasi-skeleton associated with the visible nodes $V \cup \{y_h\}$. Thus, we just need to show that such subroutine can successfully complete this procedure for a given hidden cluster.

Step 5 simply defines the sets $\tilde{V}_i$ as the visible nodes in the closure of the selected hidden cluster $C$ and Step 6 applies Theorem 17 to these sets in order to detect a rooted subtree containing a hidden root of $C$. Steps 7-14 apply Lemma 18 to find all the visible nodes connected to the hidden root of $C$. If the index set $I$ contains only $r$, we know that $T_r$ is the only rooted subtree containing $y_h$ and thus $\neg\mathcal{I}(y_h, \emptyset, y)$ for all $y \in V_r$, and $\mathcal{I}(y_h, \emptyset, y)$ for all other visible nodes $y$. If $I$ contains multiple indices, then Lemma 26 guarantees that $\neg\mathcal{I}(y_h, \emptyset, y)$ for all $y \in W$, and $\mathcal{I}(y_h, \emptyset, y)$ for all other visible nodes $y$. These last observations are at the core of Steps 15-18.

Observe that the descendants of $y_h$ in the closure of $C$ which are not directly linked to $y_h$ are the nodes in $W \cap \overline{W}$. Steps 19-23 link $y_h$ to these nodes introducing some fictitious hidden clusters. These clusters are just instrumental for the application of the merging algorithm at Step 25. Steps 25-26 merge these fictitious hidden clusters when appropriate as shown in Lemma 27 and they also update the structure of the rooted subtrees containing $y_h$ accordingly. Step 28 merges these hidden clusters using the HCMA considering all the rooted subtrees now that the node $y_h$ can be treated as visible. $\square$

## 5.6 Task 5: Obtain the pattern of the latent polytree from the recovered quasi-skeleton of the latent polytree (recover type-II hidden nodes and edge orientations)

### 5.6.1 Proof of Theorem 20

*Proof.* Steps 1-4 are an implementation of the GPT algorithm for the orientation of edges in a polytree. GPT algorithm tests two nodes $y_i$ and $y_j$ on a path of the form $y_i - y_k - y_j$. Thus, all these tests are local in the sense that they are always performed on paths of length 2 in the skeleton of the polytree. However, HRRA performs these tests on paths of length 2 on the quasi-skeleton of the polytree. If the path of length 2 on the quasi-skeleton is the same path of length 2 on the skeleton, HRRA orients the edge the same way the GPT algorithm does. The only difference arises on paths of length 2 in the quasi-skeleton which are not actual paths in the skeleton. This only occurs in situations where a Type-II hidden node is involved on the path.

There are only two possible scenarios when testing the independence statements $\mathcal{I}(y_i, \emptyset, y_j)$ or $\neg\mathcal{I}(y_i, \emptyset, y_j)$ on a path of the form $y_i - y_k - y_j$ in the quasi-skeleton of a minimal latent polytree, as depicted in Figures 6 and 7.

Figure 6: Quasi-skeleton of a rooted tree with one undiscovered Type-II hidden node (a), the detection of a conflict on the orientation of the edge $y_k - y_j$ (b), and discovery of a Type-II hidden node (c).

Figure 7: Quasi-skeleton of a rooted tree with two undiscovered Type-II hidden nodes (a), the detection of a conflict on the orientation of the edges $y_i - y_k$ and $y_k - y_j$ (b), and discovery of two Type-II hidden nodes (c).

The first scenario occurs when we have the path $y_i - y_k - y_j - y_\ell$ on the quasi-skeleton, as in Figure 6 (a). In this case, there is a yet undetected Type-II hidden node between the nodes $y_k$ and $y_j$, and the node $y_\ell$ is a parent of the node $y_j$. In this scenario, we have that $\mathcal{I}(y_i, \emptyset, y_j)$ holds giving the orientations $y_i \rightarrow y_k \leftarrow y_j$. However, because of the Type-II hidden node between the nodes $y_k$ and $y_j$ we also have $\mathcal{I}(y_k, \emptyset, y_\ell)$ implying the orientation $y_k \rightarrow y_j \leftarrow y_\ell$, as in Figure 6 (b). Thus in this scenario, the presence of the undetected Type-II hidden node is discovered from the double orientation of the edge $y_k - y_j$, as depicted in Figure 6 (c).

The second scenario occurs when we have the path $y_g - y_i - y_k - y_j - y_\ell$ in the quasi-skeleton, as in Figure 7 (a). In this case, there are two yet undetected Type-II hidden nodes: one between the nodes $y_i$ and $y_k$, and one between the nodes $y_k$ and $y_j$. Following the same reasoning as in the previous scenario, the double orientation of the edges $y_i - y_k$ and $y_k - y_j$ reveals the presence of two Type-II hidden nodes, as depicted in Figures 7 (b)-(c). $\square$

## 5.7 Conclusions and Discussion: the meaning of "minimality" for a latent polytree

In this section we show that if a latent polytree $P_\ell$ is not minimal, then there exists another latent polytree with a smaller number of hidden nodes which has the same independence relations among the visible nodes. In other words, if a latent polytree is not minimal, then there exists at least one hidden node $y_h$ that does not satisfy the minimality conditions (see the degree conditions of Definition 8). The proof of such a statement is done by considering various scenarios for such a node.

1. Case I: If $\deg_{\vec{P}_\ell}(y_h) = 1$, then this hidden node can be immediately marginalized from the factorization of the joint probability distribution to obtain an equivalent factorization where $y_h$ is not present. Indeed, if $\deg^-_{P_\ell}(y_h) = 1$, then let $y_p$ be the only parent of the node $y_h$, as depicted in Figure 8 (a). Then the factor $\mathbb{P}(y_h \mid y_p)$ disappears from the factorization of the joint probability distributions by integrating over $y_h$. Instead, if $\deg^+_{P_\ell}(y_h) = 1$, then let $y_c$ be the only child of the node $y_h$, as depicted in Figure 8 (b). Then the factor $\mathbb{P}(y_c \mid y_h) \, \mathbb{P}(y_h)$ disappears from the factorization of the joint probability distributions, again, by integrating over $y_h$.

(a)                (b)

Figure 8: A hidden node $y_h$ where $\deg(y_h) = \deg^-(y_h) = 1$ (a), and a hidden node $y_h$ where $\deg(y_h) = \deg^+(y_h) = 1$ (b).

2. Case II: If $y_h$ has a single hidden parent $y_p$ and multiple children $y_{c_1}, y_{c_2}, ..., y_{c_{n_c}}$, as depicted in Figure 9 (a), then there exists a factor in the factorization of the joint probability distribution where $y_h$ can be marginalized as follows

$$\prod_{i=1}^{n_c} \mathbb{P}(y_{c_i} \mid y_h, p^{(c_i)}) \, \mathbb{P}(y_h \mid y_p) \prod_{j=1}^{n_p} \mathbb{P}(y_{c_j} \mid y_p) \, \mathbb{P}(y_p \mid g) =$$

$$\prod_{i=1}^{n_c} \mathbb{P}(y_{c_i} \mid y_p, p^{(c_i)}) \prod_{j=1}^{n_p} \mathbb{P}(y_{c_j} \mid y_p) \, \mathbb{P}(y_p \mid g) \tag{3}$$

where $p^{(c_i)}$ are the parents of $y_{c_i}$ other than $y_h$ for $i = 1, ..., n_c$, $c_j$ are the children of $y_p$ other than $y_h$ for $j = 1, ..., n_p$ and $g$ are the parents of $y_p$, as depicted in Figure 9 (b).

(a)                (b)

Figure 9: A hidden node $y_h$ which has a single hidden parent $y_p$ and multiple children $y_{c_1}, y_{c_2}, ..., y_{c_{n_c}}$ (a), and the case where the hidden node $y_h$ is marginalized (b).

3. Case III: If $y_h$ is a hidden root with exactly two children $y_{c_1}$ and $y_{c_2}$ and at least one of its children has no other parent (without loss of generality say $y_{c_1}$), as depicted in Figure 10 (a), then in the factorization of the joint probability distribution we find a factor of the following form

$$\prod_{i=1} \mathbb{P}(y_h) \, \mathbb{P}(y_{c_1} \mid y_h) \, \mathbb{P}(y_{c_2} \mid y_h, p) \tag{4}$$

where $p$ are the parents of $y_{c_2}$ other than $y_h$, as depicted in Figure 10 (b).

Figure 10: A hidden node $y_h$ which is a root with exactly two children $y_{c_1}$ and $y_{c_2}$ and at least one of its children has no other parent (without loss of generality, say $y_{c_1}$) (a), and the case where the hidden node $y_h$ is marginalized.

By applying the Bayes' theorem, we have

$$\mathbb{P}(y_h)\, \mathbb{P}(y_{c_1} \mid y_h)\, \mathbb{P}(y_{c_2} \mid y_h, p) = \mathbb{P}(y_h \mid y_{c_1})\, \mathbb{P}(y_{c_1})\, \mathbb{P}(y_{c_2} \mid y_h, p) \tag{5}$$

and then by marginalizing over $y_h$ we obtain the following factor of the joint probability distribution

$$\mathbb{P}(y_{c_1})\, \mathbb{P}(y_{c_2} \mid y_{c_1}, p). \tag{6}$$

In all of these scenarios, one hidden node has been marginalized from the factorization of the joint probability distribution of the random variables leading to a factorization equivalent to the original one, but with fewer number of hidden nodes. In all other scenarios, the factorization is instead associated with a polytree which meets the definition of minimality of a latent polytree.