[Reviews · NeurIPS 2019]

Reviewer 1



Learning causal structures with latent variables is a major challenge. This paper takes a shot at one of the simplest cases, polytree causal networks. While this is a limited special case, the ideas and methods may be useful more generally. It is interesting that the method needs only second and third order statistics of the observed variables. The paper would benefit from greater clarity in several areas. The paper defines of a quasi-skeleton and a collapsed representation, but I don't see a definition of a collapsed quasi-skeleton. Step 3 of the process in Section 3 should be worded more carefully. First, as noted above, collapsed quasi-skeletons are not defined. Second, what does it mean for the collapsed quasi-skeletons to "partially overlap"? The collapsed representation replaces each hidden cluster with a single hidden variable; so what does it mean for two single hidden variables to "partially overlap"? An example of partial overlap might be helpful; none of your quasi-skeleton examples have any overlap in their hidden clusters. Your supplemental material gives conditions required for this to happen, but there are no examples. This makes the algorithm hard to understand. The authors have already noted in the supplemental material that Figure 1(b) needed to include orientations on the edges. This confused me because I read the paper prior to the supplemental material. Thanks also for pointing out the errors in HRRA. The abstract could be greatly improved. The first sentence says that an approach is given by a formulation. This is nonsense! An ancestral graph is a mathematical object; what is the "ancestral graph approach"? Near the end of a very difficult to follow abstract, you finally say what the paper actually is about, but don't relate it back to the problems you have been discussing with "the ancestral graph approach" (whatever that is). Update: I thank the authors for their comments, which have satisfied me.

Reviewer 2



The paper considers learning a specific type of latent variable model from conditional independence relations, a latent polytree network. The paper shows that when the underlying network has a tree structure and when certain degree conditions are met for the latent variables, the latent variable model can be exactly recovered. They show the degree conditions are complete in the sense that when they are not met, there is a polytree network with fewer latent variables which satisfy the same conditional independence relations. An algorithm is provided for learning latent polytrees when these conditions are met. In general the paper is well written. The results are novel, sound, and theoretically interesting. The paper doesn't provide much motivation for the approach so it's difficult to gauge the significance. It's not completely obvious what a realistic scenario would be where you can assume both a tree structure and the given degree conditions on the latent nodes, but don't already understand the exact latent structure. No motivating examples were provided and there are no empirical results.

Reviewer 3



The text is very well-written; the algorithm is a bit involved (as it tests several cases), but a careful reader can follow all steps provided. The class of learnable polytrees is somewhat very limited; in particular, the assumption that the number of hidden variables is known limits the applicability of the method (from a machine learning perspective). Yet I find the work an important contribution for the field, and can certainly find application in some specific domain.

[Author Response · NeurIPS 2019]

We would like to thank the three reviewers for their time and valuable suggestions. We are also glad that all reviewers have found the results of this paper significant.

---

**Response to Reviewer** #2**:** We agree that polytrees can still be considered a limited class of networks, but this class is more general compared to rooted tree models since polytrees can model fusion of information from independent sources as opposed to rooted trees. There exist a large number of articles in the literature that support the importance of studying tree structured networks since they have been shown to be useful in practice, often as an approximation of more complex networks as in [9, 10] and also [M. B. Eisen, et al., PNAS, vol. 95, no. 25, pp. 14863–14868, 1998].

We are very glad that Reviewer #2 has highlighted in his/her comments that our method requires only statistics of the second and third order and found it as an interesting property.

Reviewer #2 correctly points out that the collapsed quasi-skeleton is not formally defined. The collapsed quasi-skeleton is the collapsed representation of the quasi-skeleton of a graph. We didn't provide a formal definition because we assumed the expression to be self-explanatory. However, we gladly accept the reviewer's suggestion of adding a formal definition for this concept to improve clarity.

We also agree with the reviewer that the wording of step 3 of the algorithm can definitely be improved. We would like to clarify here, though, that by saying "the hidden clusters partially overlap" we mean that the hidden clusters in each rooted subtree have at least one hidden node in common (as mentioned in Section 3.3). There are examples of these situations in the article. For example, the polytree of Figure 3 demonstrates an occurrence of overlap between hidden clusters: the hidden node $y_{h_6}$ is shared between the hidden cluster of the rooted subtree with root $y_{h_3}$ and the hidden cluster of the rooted subtree with root $y_1$. In case of acceptance of the paper, we will make sure to include additional examples, to make the article easier to read.

We will also definitely work on improving the clarity of the overall article, of course including a rewording of the sentence which mentions "ancestral graphs" in the abstract, and adding the orientations in Figure 1(b).

---

**Response to Reviewer** #3**:** We think that the overall relevance of the article comes from the combination of the positive result ("if the degree assumptions on the hidden nodes are verified, then we can successfully recover the model") with the negative result ("if the degree assumptions are not verified, then there is a polytree model with the same observable nodes and fewer hidden nodes"). When the degree assumptions are not met, our method selects a polytree with a minimal number of hidden nodes. In other words, following a form of Occam's razor principle, our algorithm always introduces the least number of unobserved nodes to explain the observations. Thus, the theoretical relevance of the results is that either the network is recovered correctly or the simplest network (in the sense of number of hidden nodes) that explains the data is selected. We find a property of this kind quite relevant in application scenarios, too, given that Occam's razor is arguably one of the cardinal principles in all sciences. Indeed, another consequence of the negative result is that, with no additional information/observations, no test can be constructed to determine if the degree conditions are met or not.

However, there are also practical situations where the degree conditions are automatically guaranteed: for example in phylogenetic trees when all the leaves (i.e., current species) are observed [P. Erdos, et al., "A few logs suffice to build (almost) all trees", 1999].

In case of acceptance of the article, these points will be incorporated in the abstract and introduction in order to provide more motivation and context for our approach. We will also follow Reviewer #3's other recommendation to introduce an empirical validation of our method.

---

**Response to Reviewer** #5**:** We thank Reviewer #5 for the positive comments on the importance of this contribution. However, we would like to clarify that the proposed algorithm does not require to know the number of hidden nodes in advance. The only information required for the learning process is the observations of the observable nodes. The exact number and location of the hidden nodes (no matter how many hidden nodes exist in the network) is inferred by the algorithm under the assumption that certain degree conditions are satisfied on each hidden node. When these degree conditions are not verified for at least one hidden node, we show that there exists another polytree with the same observable nodes and fewer hidden nodes. Thus, our algorithm either recovers the polytree correctly or selects the simplest polytree (in the sense of number of hidden nodes) that explains the observed data.

We are more than happy to follow Reviewer #5's suggestion (shared also by Reviewer #3) to include an empirical validation of our technique.

---

[Meta-Review · NeurIPS 2019]

After a discussion among reviewers, this submission has been considered a solid theoretical contribution to the specific problem of learning polytree networks (with latent variables). It is important to understand what can be learned. The work also proposes algorithms to recover the network under some conditions. On the downside, there is no good motivations and no evaluation of any kind. The latter is hard to be resolved quickly, but an effort could be put in improving the motivations and potential impact.